# Climate-based modelling and forecasting of dengue in three endemic departments of Peru

**Cathal Mills**[1,2]*, **Christl A. Donnelly**[1,2]

**1** Department of Statistics, University of Oxford, Oxford, United Kingdom, **2** Pandemic Sciences Institute, University of Oxford, Oxford, United Kingdom

* cathal.mills@linacre.ox.ac.uk

**Data Availability Statement:** The dengue incidence surveillance data are publicly available from the National Centre for Epidemiology, Disease Prevention and Control (Peru CDC) in Peru's Ministry of Health at https://www.dge.gob.pe/

## Abstract

Amid profound climate change, incidence of dengue continues to rise and expand in distribution across the world. Here, we analysed dengue in three coastal departments of Peru which have recently experienced public health emergencies during the worst dengue crises in Latin American history. We developed a climate-based spatiotemporal modelling framework to model monthly incidence of new dengue cases in Piura, Tumbes, and Lambayeque over 140 months from 2010 to 2021. The framework enabled accurate description of in-sample and out-of-sample dengue incidence trends across the departments, as well as the characterisation of the timing, structure, and intensity of climatic relationships with human dengue incidence. In terms of dengue incidence rate (DIR) risk factors, we inferred nonlinear and delayed effects of greater monthly mean maximum temperatures, extreme precipitation, sustained drought conditions, and extremes of a Peruvian-specific indicator of the El Niño Southern Oscillation. Building on our model-based understanding of climatic influences, we performed climate-model-based forecasting of dengue incidence across 2018 to 2021 with a forecast horizon of one month. Our framework enabled representative, reliable forecasts of future dengue outbreaks, including correct classification of 100% of all future outbreaks with DIR $\geq$ 50 (or 150) per 100,000, whilst retaining relatively low probability of 0.12 (0.05) for false alarms. Therefore, our model framework and analysis may be used by public health authorities to i) understand climatic drivers of dengue incidence, and ii) alongside our forecasts, to mitigate impacts of dengue outbreaks and potential public health emergencies by informing early warning systems and deployment of vector control resources.

## Author summary

Dengue fever is a mosquito-borne infectious disease. It is a growing, substantial threat to global public health as rising climatic variability drives the expansion and increase in incidence of the disease. In the northern coastal regions of Peru, dengue is a substantial public health burden with major crises instigated by severe, recent dengue outbreaks. Here,

salasituacional. Mid-year population estimates are made available by the National Institute of Statistics and Information of Peru at https://www.inei.gob.pe/media/MenuRecursivo/indices_tematicos/proy_04.xls. The WorldClim monthly historical climate data is available at https://www.worldclim.org/. SPI-6 data from the European Drought Observatiory is available at https://jeodpp.jrc.ec.europa.eu/. The ENSO indices of the ONI and ICEN are available respectively from the NOAA (https://origin.cpc.ncep.noaa.gov/) and Geophysical Institute of Peru (http://met.igp.gob.pe). All code and data used in our analysis is available at https://github.com/cathalmills/peru_dengue/.

**Funding:** C.M is supported by a studentship from the UK's Engineering and Physical Sciences Research Council. C.A.D. is supported by the UK National Institute for Health Research Health Protection Research Unit (NIHR HPRU) in Emerging and Zoonotic Infections in partnership with Public Health England (PHE), Grant Number: HPRU200907. The funders had no role in study design, data collection and analysis, decision to publish, or preparation of the manuscript.

**Competing interests:** The authors have declared that no competing interests exist.

focusing on three Peruvian departments of Piura, Tumbes, and Lambayeque, we describe the extent to which current and past recent climatic conditions, such as temperature, precipitation, and sea-surface temperatures, shape the patterns in reported dengue cases. Furthermore, we display how understanding such relationships with climatic conditions can be used to model historical trends in dengue cases, which can be helpful for addressing incomplete dengue surveillance. Finally, the study demonstrates how climate-model-based forecasting of dengue cases can be employed for reliable prediction of potential future outbreaks up to one month ahead of time. The insights can be used by public health authorities before and during dengue outbreaks to potentially reduce the morbidity, mortality, and strains on health services by informing public health policies such as information campaigns, early warning systems, and targeted mosquito control interventions.

# 1 Introduction

In recent years, various analyses globally have quantified complex interactions between climate variability and the dynamics of vector-borne infectious diseases, such as dengue, malaria, Zika, and chikungunya [1–3]. Dengue virus (DENV) is an example of a flavivirus that is transmitted to humans by infected *Aedes* mosquitoes (primarily *Ae. aegypti*). Endemic in over 100 countries, dengue continues to pose a substantial public health concern worldwide, particularly in tropical and subtropical regions where the *Ae. aegypti* mosquito thrives. Incidence of dengue cases has risen significantly in recent decades, alongside an expanding geographical distribution that is attributed to factors that regulate transmission dynamics such as urbanisation, globalisation, increased travel, inadequate vector control measures, and climate change [4, 5].

Here, we quantify the important role played by the climate in the timing and intensity of reported dengue transmission across endemic regions of Peru. In many other nations, complex, non-linear associations are well-established between incidence of dengue cases and climatic variables such as temperature and precipitation [1, 6]. Climatic variables directly influence the abundance, distribution, and activity of *Ae. aegypti* mosquitoes, as well as the replication and survival of DENV within the mosquito vector. In particular, warmer temperatures accelerate mosquito development, shorten their incubation period, and increase their biting rates. Meanwhile, rainfall generally provides breeding sites for *Ae. aegypti* mosquitoes, promoting their population growth [3, 7–11].

Extreme climatic events, such as drought and heavy rainfall, have complex, spatially-dependent relationships with dengue transmission. Whilst drought can reduce dengue transmission by limiting mosquito breeding sites and mosquito populations, water scarcity can conversely prompt individuals to store water in containers that serve as mosquito breeding sites, thus producing conditions that favour the survival and proliferation of mosquitoes [1, 12–14]. Similarly, during periods of intense rainfall, water typically accumulates in containers such as flower pots and water storage containers, thus providing ample breeding grounds for mosquitoes [15–17]. In terms of human factors, heavy rainfall can result in the overflow of sewage systems, increasing the risk of contamination and facilitating the spread of dengue.

In Peru, dengue has long been a significant public health concern, characterised by periodic outbreaks and year-round transmission. In recent years, there have been major dengue outbreaks where annual reported incidence has reached record levels within the nation. The studied departments of Piura, Tumbes, and Lambayeque are located in the northern coastal region of Peru, and were chosen in this study due to their presence at the epicentre of recent severe dengue outbreaks [18]. We were interested in quantifying climatic influences on dengue

epidemic dynamics in the region as the three departments experience favorable environmental conditions for *Ae. aegypti* mosquito breeding due to their warm and humid climate, whilst coastal proximity increases exposure to extreme El Niño-induced climatic events. Piura, Tumbes, and Lambayeque have also faced challenges in dengue control due to population growth, sanitation infrastructure, and resources for vector control programmes.

Understanding the climatic influences on dengue transmission in the three departments is essential for development of effective strategies to prevent and control disease transmission. By characterising the relationships between climatic variables and dengue incidence, public health authorities can attain insights for targeted interventions, early warning systems, and resource allocation, thus potentially mitigating the ongoing, persistent public health impact of dengue.

## 2 Methods

The following section describes the data, study area, and modelling techniques used to model reported dengue cases in the Peruvian departments of Lambayeque, Piura, and Tumbes from 2010 to 2021.

### 2.1 Data

We obtained department-level surveillance data on reported monthly dengue cases (which includes both probable and confirmed cases across dengue serotypes) from the National Centre for Epidemiology, Disease Prevention and Control (Peru CDC) in Peru's Ministry of Health [19]. The Peru CDC define a probable dengue case for an individual who i) has febrile illness for at most seven days, ii) has two or more of a list of specified symptoms, iii) and lives in or has recently visited areas with either known dengue transmission or known *Ae. aegypti* populations. Confirmed dengue cases meet the same exposure and symptoms criteria, alongside a positive dengue test [18]. We sourced mid-year population estimates (as of 30 June) for each of the three departments from the INEI (National Institute of Statistics and Information of Peru), and for each month, we linearly interpolated the yearly population values to avoid improbable population fluctuations [20].

In terms of meteorological data, we sourced monthly precipitation and temperature data from the WorldClim 2·1 dataset, which provides open-source climate data globally in fine spatial resolution of 1km [24, 25]. We derived monthly estimates of average daily maximum temperature (˚C), average daily minimum temperature (˚C), and total precipitation (mm) for each studied department by averaging the variables across the grid cells corresponding to each department.

We used the Standardized Precipitation Index at a six-month time scale (SPI-6), a widely-used index in hydrology and climatology, to assess and monitor drought conditions based on precipitation data [26, 27]. SPI-6 measures how anomalous the accumulated precipitation over six months is compared to historical records, and accounts for both the magnitude and duration of precipitation deficits or surpluses. As SPI-6 is calculated over a medium-term accumulation period, it is an indicator of reduced stream flow and reservoir storage. An advantage of SPI-6 is that is a relative index as it represents the deviation from the average conditions specific to a location. Positive SPI-6 values indicate wetter-than-average conditions, whilst negative values indicate drier-than-average conditions. The SPI-6 data were sourced from a dataset collated by the European Drought Observatory [28].

Finally, for characterising the impacts of the El Niño Southern Oscillation (ENSO) on human cases of dengue, we considered two climate indices of the Pacific Ocean; the El Niño Coastal Index (ICEN) and the Oceanic Niño Index (ONI). The data on the indices were sourced respectively from the Geophysical Institute of Peru (IGP) and the National Oceanic and Atmospheric Administration in the United States. First, the ICEN, maintained by the

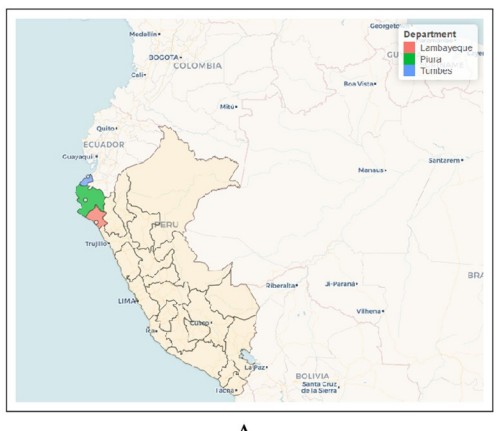

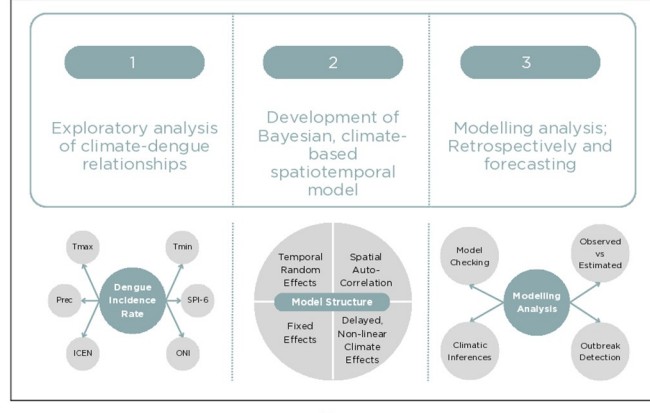

A                                                                                          B

**Fig 1. Study setting and schematic of methodology. A**: Geography of Peru (shaded amber), neighbouring countries and the Pacific Ocean. The three studied departments of Lambayeque, Piura, and Tumbes are highlighted in red, green, and blue respectively, and their capital cities are represented by filled circles. The map was created using the mapview and rgeoboundaries packages in R version 4·2·1 [21–23]. The base layer map is available at https://www.geoboundaries.org/api/current/gbOpen/PER/. **B**: Overview of the approach taken for analysis of dengue incidence in the three departments across 2010 to 2021. Details of acronyms used in the schematic and throughout the manuscript can be found in Section 1 in S1 Appendix.

IGP, is a climate monitoring index which is derived from the three-month running average of the sea surface temperature (SST) anomaly in the Niño 1+2 region and the climatology of the period between 1981 and 2010 [29, 30]. As the ICEN is a bespoke index that is specific to Peru, it is generally more representative of ENSO conditions for Peru than large-geographic-scale monitoring metrics (such as the ONI). Here, due to our geographic focus on the Peruvian coast, we concentrated on the E-Index component of the ICEN which summarises the variability associated with El Niño (warming events) and La Niña (cooling events) via anomalous surface warming in the eastern Pacific Ocean. Nevertheless, we also extracted values of the ONI, which provides a three-month rolling average of the SST anomaly in the Niño 3·4 region [31]. ONI values greater than 0·5 are indicative of El Niño events, whilst ICEN E-Index values greater than 1·7 are classified as strong El Niño events.

### 2.2 Study area

The three departments of Piura, Tumbes, and Lambayeque are located in the northern coastal region of Peru (Fig 1). As of June 2021 (the latest estimates for the studied period), the populations of Piura, Tumbes, and Lambayeque were estimated to be approximately 1,326,000, 2,077,000, and 256,000 respectively (Table B in S1 Appendix), whilst the corresponding population percentages living in urban areas were 79·3%, 93·7%, and 81·1%. The climates of each department are characterised by generally dry and hot conditions, as well as distinct dry and wet seasons. The neighbouring departments experience broadly similar climatic conditions (Fig G in S1 Appendix), and also share similarities in seasonal trends for dengue incidence rate (DIR) per 100,000 population (Fig 2 and Fig H in S1 Appendix) [55, 56]. Whilst the departments have historically differed in the average DIR observed, all three departments have experienced recent endemicity of dengue and have been afflicted by severe dengue outbreaks in 2015 and 2017 and more recently in 2023 and 2024 (which are not part of the current studied period).

### 2.3 Modelling framework

We developed a bespoke, climate-based Bayesian spatiotemporal model which was heavily shaped by region-specific knowledge of the Peruvian coastal climate and its interplay with

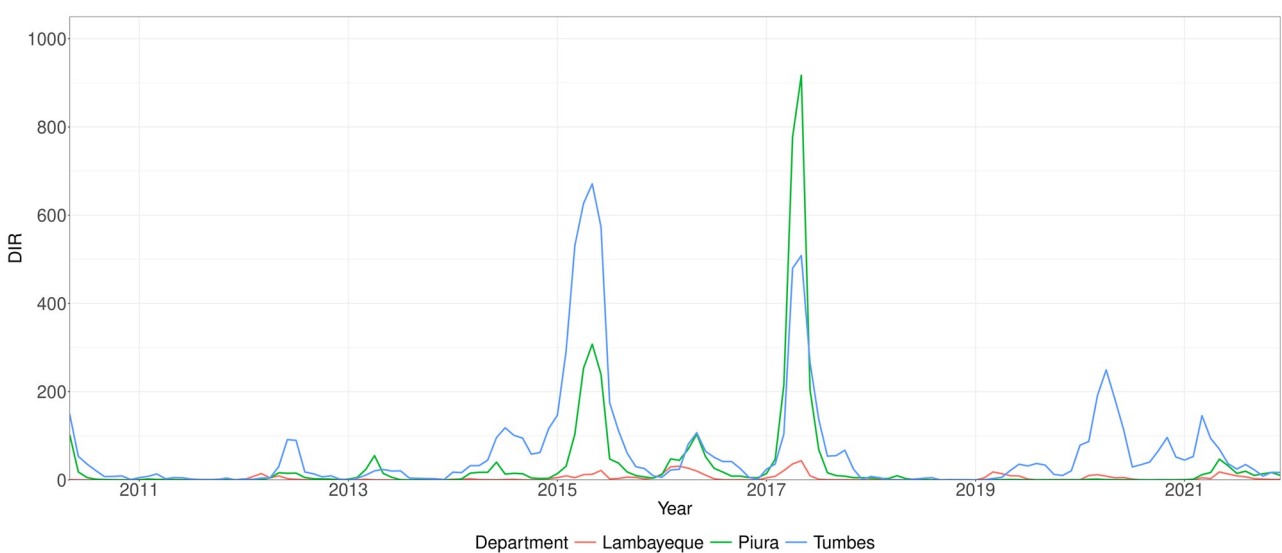

**Fig 2. Dengue incidence rate (DIR) trends over time across the departments.** The monthly DIR per 100,000 is displayed for each of the three studied departments across 2010 to 2011 inclusive. Substantial peaks in monthly DIR occurred in 2015 and 2017.

dengue incidence. Our underlying model structure was informed by previous climate-based spatiotemporal analyses of dengue incidence in other nations [1, 6]. In particular, our Bayesian hierarchical model of monthly dengue incidence allowed for temporal random effects, spatio-temporal effects (both structured and unstructured), and delayed, potentially non-linear effects of climatic variables. We employed domain knowledge and insights from our extensive exploratory analysis of the climatic influences which allowed us to identify factors specific to Piura, Tumbes, and Lambayeque, as well as initial insights into the strength and nature of the relationships between the climate and reported cases. In our exploratory analysis, we described similarities across departments in the structure and timing of relationships between climatic variables and DIRs (Figs A and G in S1 Appendix). Alongside the random effects and climatic variables, our model incorporated a common seasonality indicator for the Peruvian summer months and an innovative momentum indicator, the lagged value of the Relative Strength Index (RSI), which was inspired by analysis techniques of financial markets. The epidemiological logic behind the RSI is that it approximates the epidemic trajectory and thus aims to assist in tracking and forecasting temporal trends in incidence of new dengue [32]. Whilst a brief description of the model framework is provided here, further descriptions are provided in Section 3 of S1 Appendix.

To allow for potential overdispersion, we assumed a negative binomial likelihood for the monthly case counts of dengue $y_{it}$ for each department ($i = 1, 2, 3$) across the studied months ($t = 1, \ldots 140$):

$$
\begin{aligned}
y_{it} &\sim NegBin(\mu_{it}, \kappa) \\
\log(\mu_{it}) &= \log(P_{it}) + \log(\eta_{it}) \\
\log(\eta_{it}) &= \beta_0 + \mathbf{x_i}\boldsymbol{\beta} + \gamma_{i,m(t)} + \delta_{i,a(t)} + u_i + v_i + \sum_{l=1}^{L} bs(c_{lit}, 2, 4)
\end{aligned}
\tag{1}
$$

where the negative binomial distribution had mean parameter $\mu_{it}$ and overdispersion

parameter $\kappa$. We included a population offset $\log(P_{it})$, where $\log()$ denoted the natural logarithm, such that $\eta_{it}$ represented the DIR per 100,000 population. We then modelled the DIR with an intercept $\beta_0$, alongside fixed and random effects. $\boldsymbol{x_i}$ (with corresponding parameter vector $\boldsymbol{\beta}$) represented a matrix of fixed effects; the binary seasonality variable (1 for summer months of December to April, and 0 otherwise) common across departments and the department-specific RSI variable lagged by one month. To account for cyclical patterns of monthly temporal dependence, we used monthly random effects ($\gamma_{i,m(t)}$) specific to each department, which were specified by cyclic RW(1) (random walk of order one) prior distributions. Furthermore, to account for unobservable year-to-year heterogeneities, such as changes to reporting rates, vector control programmes, or population-level immunity, we included exchangeable (Gaussian-distributed independent and identically distributed) department-specific yearly random effects $\delta_{i,a(t)}$. Sensitivity analyses revealed that pooled monthly or annual random effects were too primitive to adequately capture the departments' monthly and annual variations in DIRs respectively. $u_i$ and $v_i$ represented the structured and unstructured spatial random effects respectively of our a reparameterisation of the traditional BYM (Besag, York and Mollié) model, called the BYM2 model, which implements Penalised Complexity prior distributions to enable scaling and provide interpretable parameters [33, 34]. Finally, to allow for varying lagged and potentially non-linear effects of maximum temperature, precipitation, SPI-6, and ICEN, we used distributed lag non-linear models (DLNMs) for our four ($L = 4$) climatic covariates $c_{lit}$ via quadratic B-splines $bs(c_{lit}, 2, 4)$ with two degrees of freedom in the exposure-response dimension, a maximum lag of four months, and an internal knot at two months in the lag-response dimension (further details in S1 Appendix Section 3.5) [35].

The model was implemented using Integrated Nested Laplace Approximation (INLA) in R version 4·2·1 [23, 36]. In addition to assessments of model structure, we specified our final model by analysing performance across diagnostics such as Deviance Information Criterion (DIC), cross-validated (CV) log score, Mean Absolute Error (MAE), posterior predictive checking, and Probability Integral Transform (PIT) values [37].

## 2.4 Modelling analysis

We divided our modelling analysis into three environments; i) model fitting to the entire dataset of 140 months of dengue notifications across the three departments, ii) retrospective modelling with leave-one-out cross-validation (LOOCV), and iii) model-based forecasting with a forecast horizon (or lead time) of one month.

First, our model fit to the entire dataset was the basis for drawing conclusions about the climatic relationships with new dengue cases. We substantiated conclusions by extensive sensitivity analyses where we varied the model structure (S1 Appendix Section 4.2). Second, in our LOOCV approach (more precisely a leave-one-time-point-out approach) we excluded each time point's (three) monthly observations one at a time, refitting the model to a reduced dataset of 417 observations (139 time points) and generating posterior predictive distributions for each of the three excluded observations. We assessed model performance on the out-of-sample observations via outputs such as the proportion of DIR observations within the 95% credible intervals of the posterior predictive distributions. In doing so, we (in part) externally validated our model's predictive performance in synthetic out-of-sample environments to reduce the likelihood of overfitting. In the LOOCV environment, we further generated posterior probabilities that a monthly DIR exceeded specific thresholds (such as 50 per 100,000) by computing the proportion of each observation's posterior predictive samples which exceeded the corresponding threshold. We assessed our model's ability to accurately detect outbreaks of various sizes in the out-of-sample settings via visualisations of the posterior probabilities (compared

against the observed DIR values) and accuracy metrics for classification of outbreak events (predicted if posterior probability exceeded an outbreak classification cut-off probability, defined below). Note that our public-health-oriented exceedance thresholds (for outbreak months) were pre-specified by us to indicate different magnitudes of dengue outbreaks.

Probabilistic forecasting involved the exclusion of an individual month's data and all future data (both climatic and epidemiological) from the model fitting procedure, and then using only data with a lag of one month or more to forecast the specified individual month's DIRs. In the forecasting setting, we focused on 2018 to 2021, a period of four years (one month at a time) to ensure that we had sufficient (surveillance and climate) data for training a well-informed model for reliable forecasting. We performed similar assessments to the LOOCV environment, including posterior predictive 95% credible interval checks and outbreak posterior probabilities. For classification of outbreaks, we did not employ an arbitrary cut-off posterior probability as we used our retrospective modelling analysis to identify an appropriate cut-off probability. In particular, using only historical data from 2010 to 2017, our optimisation procedure identified a cut-off probability (for each threshold of 50 per 100,000 or 150 per 100,000) that historically maximised the percentage of true outbreaks detected, whilst simultaneously minimising the probability of false alarm. Whilst the calibrated outbreak detection performance for the LOOCV historical analysis is an upper bound on historical model performance, for the important use-case setting of real-time forecasting, the cut-off was calibrated using past data only. The procedure was equivalent to maximising the historical AUC, the area under the Receiver Operating Characteristic (ROC) curve. Thus, we ensured that our outbreak cut-off probability was informed by i) priorities of public health authorities (i.e. reliable, robust forecasting of outbreaks) and ii) historical results. Overall, the objective of the forecasting analysis was to determine whether climate-based forecasting could be useful for real-time prediction of future trends in incidence of new cases and reliable detection outbreaks of varying magnitudes one month ahead of time.

## 3 Results

### 3.1 Characterising climatic influences on incidence of new dengue cases

The exploratory cross-correlation plots (Fig A in S1 Appendix) capture the structure and intensity of the linear relationships between lagged values of climate variables and the current DIR. Here, we identified strong similarity in the nature of each department's relationships between climatic factors and DIRs. From a forecasting perspective, the climatic relationships with DIRs were generally strongest (magnitude of correlation) at lead times of between one to three months. The strength of such lagged relationships indicate potential for developing useful climate-model-based forecasts at least one month ahead of time.

Following our exploratory analysis, our Bayesian spatiotemporal model framework (described in Section 2.3), in particular the specified DLNMs, allowed us to precisely quantify the directional relationship between DIRs and individual climatic variables. We estimated these relationships in the presence of other climatic variables and after accounting for seasonality effects, momentum effects, temporal random effects, and spatial random effects.

For each climatic variable, the three-dimensional exposure-lag-response relative risk (RR) plots (Fig 3) display, for different lags and different levels of a climate variable, the estimated elevated (or reduced) level of risk of incidence of new dengue cases relative to the risk induced by the corresponding mean value of the climatic variable. First, greater mean monthly maximum temperatures (above the mean value of 28˚C) were associated with higher RR of incidence of new dengue cases, whilst lower values were associated with lower levels of RR. In terms of the cumulative risk induced by maximum temperatures (Fig I in S1 Appendix),

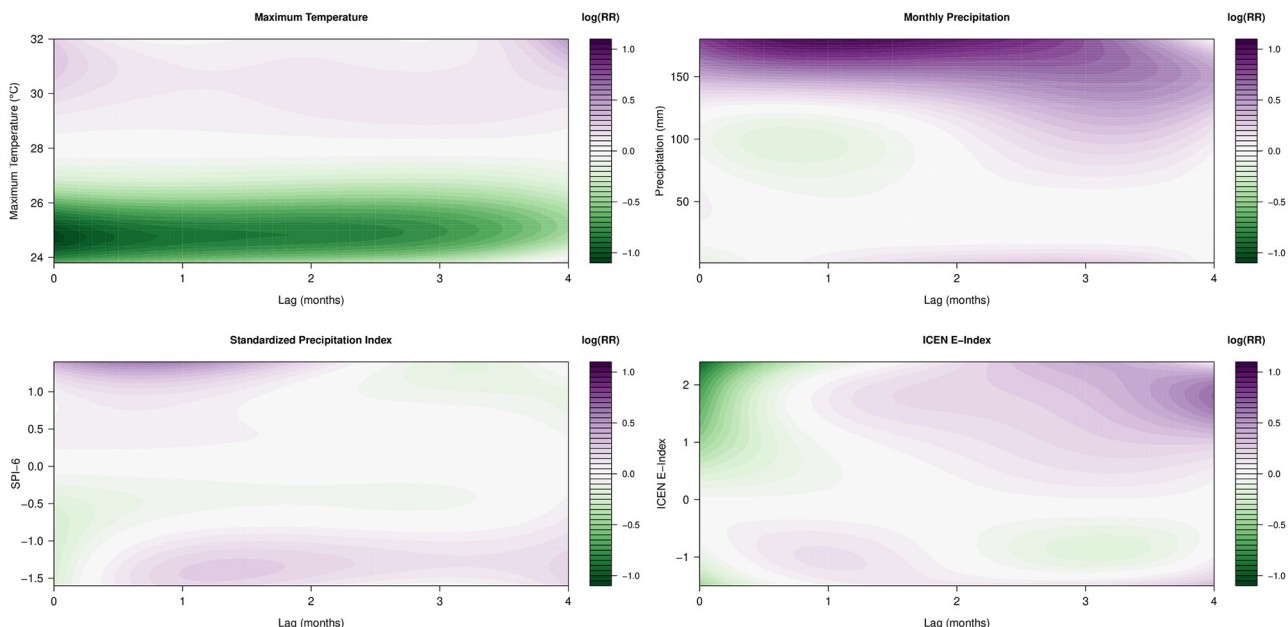

**Fig 3. Model-based exposure-lag-response relationships between climatic variables and dengue incidence rates.** Plots of relative risk (RR), on a natural logarithm scale, for the included DLNMs (Distributed Lag Non-linear Models) in our climate-based Bayesian hierarchical model for incidence (of new dengue cases) fitted to the entire period of 140 months, where RR is defined relative to the risk induced by the mean observed value of each climate variable. Log RR values greater than 0 (pink to purple) correspond to heightened RR of incidence of new dengue cases, whilst values less than 0 (green) correspond to reduced RR. The four climatic variables were included in the model framework via DLNM specifications alongside temporal random effects, spatiotemporal random effects, and fixed effects (of momentum and seasonality).

higher levels of maximum temperature (e.g. 31˚C) were consistently associated with greater cumulative risk, whilst lower maximum temperatures (e.g. 25˚C) resulted in reduced cumulative risk. Second, the exposure-lag-response RR plot corresponding to total precipitation captures that extreme wet conditions in the preceding four months were associated with elevated RR of DIR. Similarly, extreme low total precipitation values in the preceding months were found to contribute to higher RR, albeit at lower levels and only with a more substantial time lag of about one to three months. We further captured the precipitation findings in our plots of cumulative risk (Fig I in S1 Appendix), where extremities of total precipitation were associated with elevated cumulative risk, with the greatest risk occurring for extreme rainfall values over approximately three months. Third, the findings for SPI-6 were similar in terms of moderate-to-extreme rainfall and severe drought, albeit with a slightly different interpretation as SPI-6 is a cumulative and location-specific hydrological drought index which measures accumulated precipitation deficits over six months. Here, we found that lower values of this six-month-accumulated drought index (i.e. moderate-to-extreme drought), in any of the preceding one to four months, were associated with elevated RR and elevated cumulative risk, indicative of a lagged effect of drought conditions. Conversely, larger values of the SPI-6 (i.e. moderate-to-extreme precipitation) contributed to greater RR at more recent lags of zero to two months. Finally, in terms of the ICEN E-Index, we found that the greatest RR contribution, relative to the risk posed by the mean E-Index (−0·15), stemmed from IGP-classified *strong* El Niño events (greater than 1·7), followed by smaller RR contributions from *moderate* La Niña events (less than −1·4). In both cases, the lagged values of the index variable (by at least one month) were most important in terms of contributing to higher RR.

## 3.2 Climate-based retrospective modelling

Using our climate-based Bayesian spatiotemporal model, we accurately captured in-sample temporal trends in DIR from 2010 to 2021 across the three departments (Fig J in S1 Appendix), whilst also generalising well to synthetic out-of-sample environments. Performance of our model (described in Section 2.3) was appraised using a range of metrics such as DIC, CV log score, MAE, and cross-validated posterior predictive checking. The model's LOOCV posterior predictive check revealed that 94·7% of the DIR observations lay within the 95% credible intervals of our leave-one-out posterior predictive distributions. We display the out-of-sample predictive performance of our modelling framework in each department across the studied months in Fig 4, whilst model-based errors are visualised in Fig O in S1 Appendix. In general, there was a moderate-to-strong ability to track out-of-sample trends in DIR, ranging across substantial dengue outbreaks and periods of lower DIR. We note a caveat that extreme large reported DIR observations (e.g. Piura in May 2018) occasionally resulted in under-estimation of the extent of an outbreak and widening of credible intervals (see Discussion).

To further assess our model's capabilities for reliable detection of outbreaks, we produced probabilistic estimates of out-of-sample observations exceeding (pre-defined) outbreak-indicative DIR thresholds of 50 per 100,000 and 150 per 100,000 (visualised, and stratified by true outbreak months and non-outbreak months, in Figs P and Q in S1 Appendix). In terms of out-of-sample outbreak classification (Table C in S1 Appendix), for outbreaks of DIR greater than 50 per 100,000 and greater than 150 per 100,000, our model enabled detection of 93% and 100% of such events respectively, and the respective false positive/alarm rates (the proportion of incorrectly classified non-outbreaks) were 0.13 and 0.06.

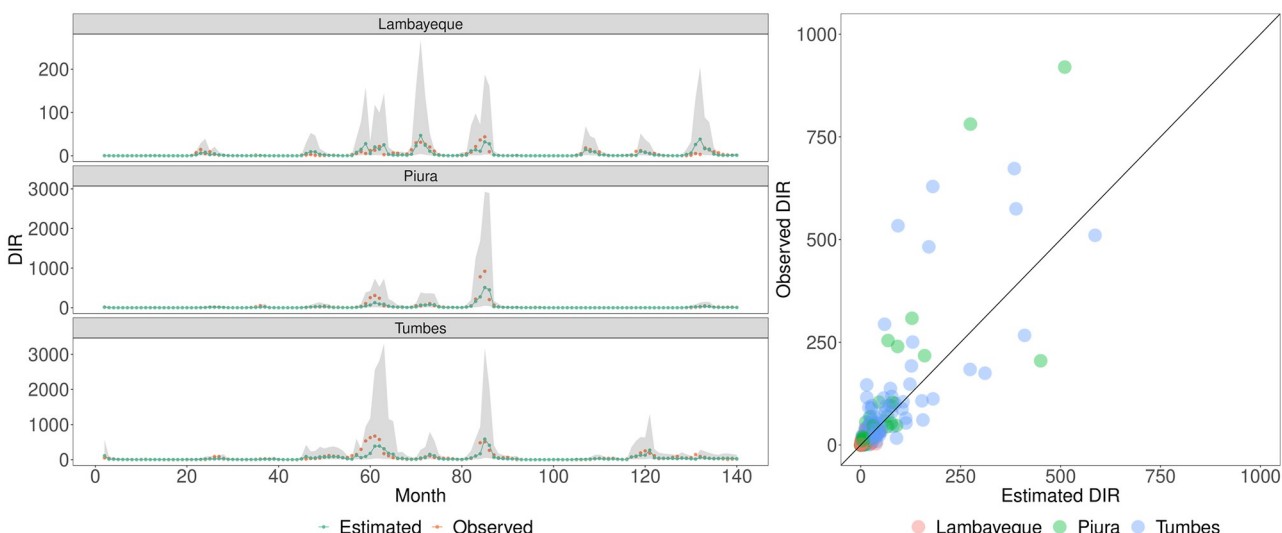

**Fig 4. Leave-one-time-point-out cross-validation predictive performance.** Left: The DIR (Dengue Incidence Rate) time series (gold) for each department is shown alongside the posterior median estimate (forest green) for each observation and the corresponding 95% credible intervals of the posterior predictive distributions (shaded grey). Estimates were obtained by refitting the model 139 times, excluding a single time-point/month (of three observations) one at a time, and estimating the posterior predictive distributions for the three omitted observations. Note that due to the presence of temporal autocorrelation terms (such as a random walk of order one prior distribution), leave-one-out predictive distributions for the observations at the first time-point are not generated. Right: The accompanying visualisation displays the observed DIR versus the corresponding estimated DIR, where the filled colours of light red, light green, and light blue represent Lambayeque, Piura, and Tumbes respectively.

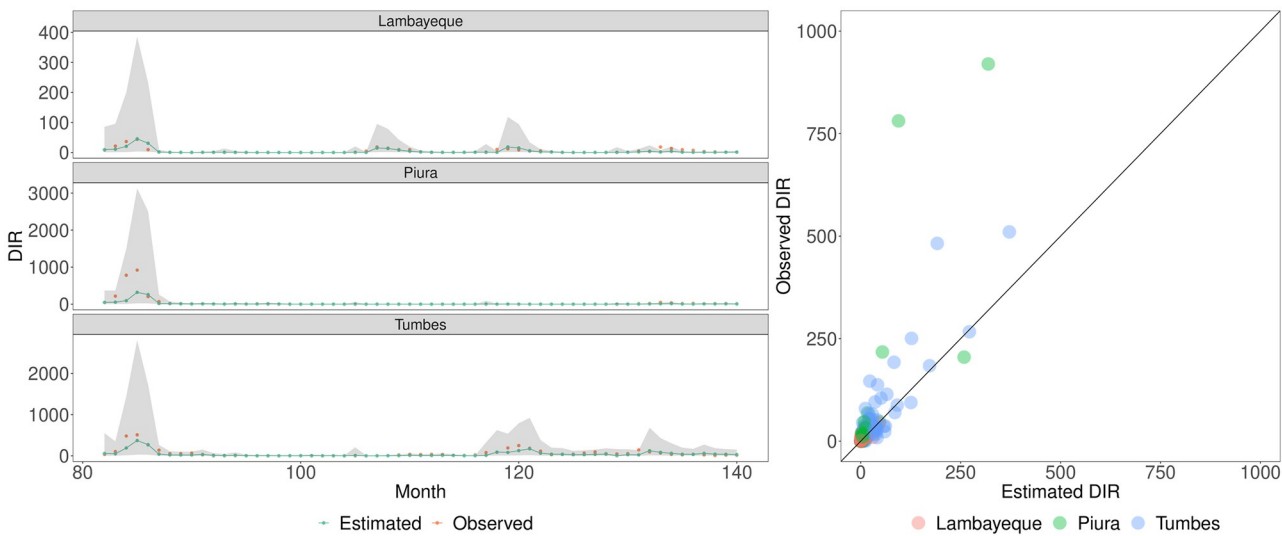

**Fig 5. Forecasting predictive performance across 2018 to 2021.** Left: The DIR (Dengue Incidence Rate) time series (gold) for each department is shown alongside the posterior median estimate (forest green) for each observation, and the corresponding 95% credible intervals of the posterior predictive distributions (shaded grey) which were obtained by fitting our model to the climatic and surveillance data up to one month preceding, and estimating the posterior predictive distributions for the next month's three observations. Right: The accompanying visualisation plots the observed DIR vs the corresponding estimated DIR, where the filled colours of light red, light green, and light blue represent Lambayeque, Piura, and Tumbes respectively.

### 3.3 Climate-model-based forecasting

The analysis above presents our estimated climatic impacts on incidence of new dengue cases and retrospective results of our climate-based spatiotemporal model. Here, focusing on a public health use case, we appraise the forecasting utility of our model with a forecast horizon of one month. Across our forecasting analysis' studied period of 2018 to 2021, our model-based forecasting of the subsequent month's DIR was relatively accurate, as displayed by close correspondence between the observed DIR time series and forecasted values. Fig 5 visualises the association between the observed DIR values and the model-based forecasted values, whilst Fig R in S1 Appendix captures the posterior median estimates of absolute errors. Specifically, 93·9% of DIR observations lay within the 95% credible intervals of the posterior predictive distributions. In general, large peaks in DIR values were correctly forecasted one month ahead, albeit with a caveat that extreme DIR observations in May to June of 2018 were under-estimated by posterior median estimates but lay within the relatively wide 95% credible intervals.

In terms of quantifying the anticipation of future outbreaks (Table 1, Fig 6) using our model (alongside historically calibrated outbreak cut-off probabilities), we correctly forecasted the occurrence of 100% of future dengue outbreaks where DIR exceeded 50 per 100,000 or 150 per 100,000. With respect to the reliability of the forecasts for outbreaks, false positive rates were 0.12 and 0.05 respectively, thus indicating relatively low probabilities of false alarms.

## 4 Discussion

In our analysis, we developed a bespoke climate-based Bayesian hierarchical model which retrospectively modelled and prospectively forecasted spatiotemporal trends in dengue incidence across three Peruvian coastal departments. Retrospectively, across the twelve studied years from 2010 to 2021, we characterise our model's predictive performance by representative DIR estimates for individual observations that were excluded from the model-fitting process.

**Table 1. Summary statistics for forecasting outbreak months across 2018 to 2021.** DIR (Dengue Incidence Rate) observations (*n* = 180) were classified one month ahead of time as being predicted true outbreaks with DIR exceeding 50 (or 150) per 100,000 if the posterior probability of DIR exceeding 50 (or 150) was greater than a cut-off probability of 0·21 (or 0·13). *True Positive* measures the hit rate, or equivalently the proportion of true outbreaks correctly detected, whilst *False Positive* measures the proportion of non-outbreaks incorrectly classified as outbreaks. *Accuracy* represents the proportion of observations whose outbreak classification matched the true observed state (such as a predicted outbreak coinciding with an outbreak). *Precision* measures the proportion of predicted outbreaks which were true outbreaks. Finally, the *AUC* is the area under the Receiver Operating Characteristic (ROC) curve, is used as a measure of our model's skill for distinguishing between outbreaks. The cut-off probabilities were calibrated using historical data (prior to 2018) and thus, outbreak forecasting model performance is reflective of a realistic application of our framework by authorities. 95% CIs are the corresponding 95% confidence intervals [57, 58]. Further analyses of outbreak classification performance were performed (see Table D in S1 Appendix).

| Outbreak Threshold | True Positive | False Positive | Accuracy | Precision | AUC |
|---|---|---|---|---|---|
| **DIR ≥ 50** (95% CI) | 1·00 (0·87, 1·00) | 0·12 (0·07, 0·18) | 0·90 (0·85, 0·94) | 0·59 (0·43, 0·74) | 0·94 (0·77, 0·93) |
| **DIR ≥ 150** 95% CI | 1·00 (0·69, 1·00) | 0·05 (0·02, 0·10) | 0·95 (0·91, 0·98) | 0·53 (0·35, 0·87) | 0·97 (0·96, 0·99) |

Similarly, we produced reliable outbreak detection estimates in synthetic out-of-sample environments, including correct identification of 100% of out-of-sample outbreaks with DIR greater than 150 per 100,000, and a low false positive rate of 0.06. In our model-based analysis, which leveraged existing research [1, 6] and our earlier exploratory analysis, we uncovered the following setting-specific inferences about the strength, timing, and structure of climatic

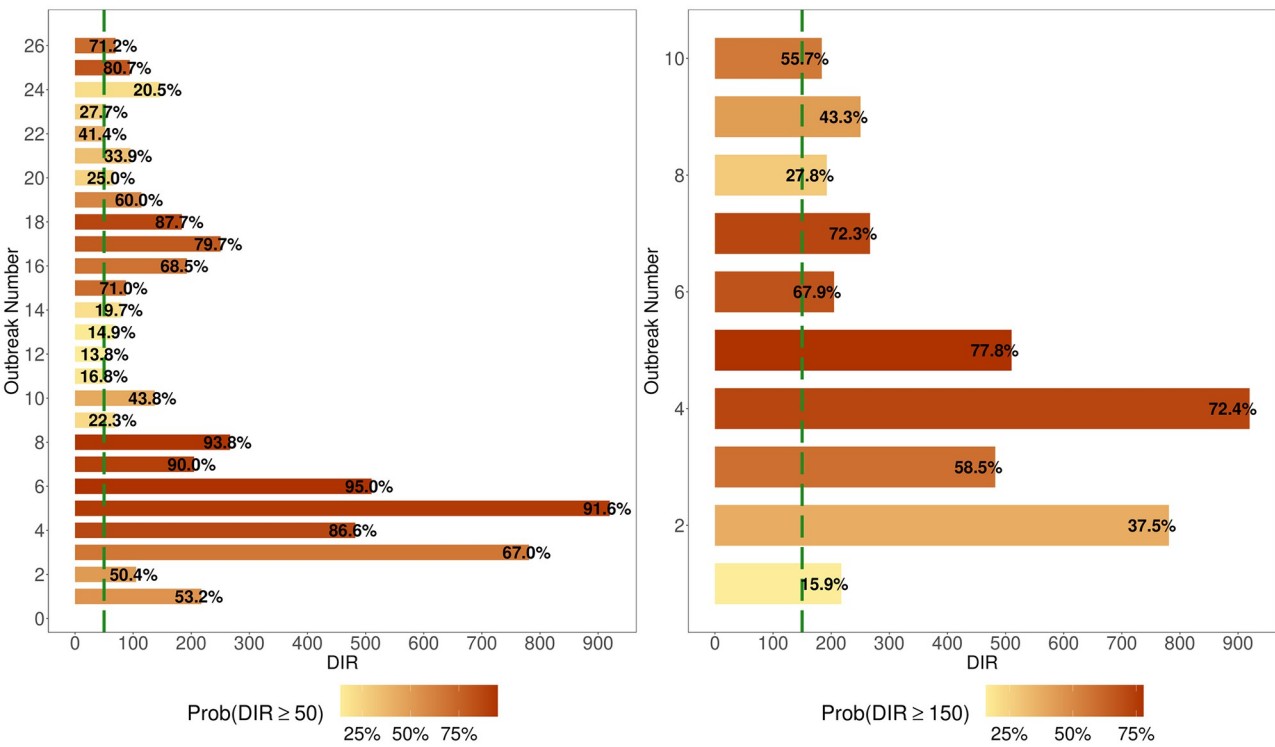

**Fig 6. Forecasting outbreak detection across 2018 to 2021.** Among the observations with DIR (Dengue Incidence Rate) greater than 50 (left) and DIR greater than 150 (right) per 100,000, the plots depict posterior probabilities of forecasted DIR exceeding thresholds (green) of 50 per 100,000 (left) and 150 per 100,000 (right). The plots capture the model's sensitivity in forecasting substantial dengue outbreaks one month in advance. To also visually assess the reliability of forecasted outbreaks and ensure a representative picture of our model's future outbreak detection capabilities, Fig S in S1 Appendix is the analogous visualisation for the corresponding posterior probabilities of the observations with observed DIR less than the thresholds of 50 and 150.

influences on incidence of new dengue cases. Our model-based inferences were substantiated by extensive model checking across diagnostics and sensitivity analyses (which varied the model structure).

We found that greater maximum temperatures (between 29°C and 31°C) promoted heightened risk of incidence of new dengue cases, which is aligned with many clinical and observational studies of the *Ae. aegypti* vector and DENV. For instance, the optimal temperature for dengue transmission has been estimated to be approximately 29.3°C, whilst in Singapore, the overall cumulative risk (induced by maximum temperatures) for incidence of new dengue cases was estimated to have peaked at 31°C [3, 8, 9]. Here, our estimated elevated risk induced by greater maximum temperatures may be due to any of a myriad of factors including accelerated mosquito larvae development, faster reproduction rates, greater mosquito survival rates, reduced extrinsic incubation period of the dengue virus, heightened mosquito biting activity, and/or faster viral replication within the vector [7, 10, 11, 38]. The estimated risk contributions from greater maximum temperatures at different time lags (across the four months) may be due to the time required for one or more of the following; temperatures to influence both larvae development and oviposition (with maximum temperature effects on oviposition activity in Ecuador estimated to be most significant at a six-week time lag), the mosquito life cycle (development occurs in four stages of egg, larva, pupa, and adult, which is estimated to be around two weeks), the incubation period of DENV (time between mosquito bite and the symptom onset of the infected individual, which is estimated to be three to ten days), and/or the time until dengue case notification [10, 39].

We obtained similar findings for extreme precipitation events, especially heavy rainfall (reflected in our precipitation and SPI-6 variables). Heavy rainfall influences mosquito population dynamics by creating additional breeding habitats via the accumulation of stagnant water in settings such as water storage containers, discarded tires, and flower pots. Thus, ample breeding sites are created for *Ae. aegypti* to lay their eggs in stagnant water, and the mosquito population (in size and density) can increase, thus providing greater opportunity for mosquito-to-human contact. The effects of precipitation on the abundance of *Ae. aegypti* specifically are pronounced due to the preference of *Ae. aegypti* to oviposit in such artificial water storage containers (as opposed to vegetation) which can become feasible egg-laying sites due to precipitation [15–17]. Water storage practices in our studied departments, all of which are highly urbanised, have been influenced by extreme drought conditions and limits on daily water supplies which is amplified by the distinct dry and wet seasons. More generally, the role played by water storage containers in the generation of oviposition sites has been estimated to be dominant in dry seasons whereas solid waste (e.g. plastics) generally provide the sites in rainy seasons [40]. Our conclusion about the heightened risk posed by lagged extreme precipitation is in agreement with existing findings of stronger relationships for lagged values. For instance, aligning with our exploratory analysis, the strongest correlation with incidence of new dengue cases was estimated for two-month-lagged precipitation in Mexico and for one-to-two month lagged precipitation in Barbados, whilst in China, the strongest relationship was for lagged precipitation of three to four months [6, 41, 42].

Using our drought indicator (SPI-6), we estimated strong relationships between sustained drought conditions (accumulated precipitation deficits) and subsequent (lagged) heightened risk of incidence of new dengue cases. Our findings are similar to dengue research in other regions globally where similar lagged associations have been observed with drought periods [1, 13, 14]. The complex drought-induced risk may be due to changes to water storage and conservation patterns as households increase water storage in containers around the home, thus creating potential egg-laying sites for female *Ae. aegypti* [43, 44]. As there may be fewer available breeding habitats, concentrated breeding may occur in the limited remaining water

bodies. Here, lagged values of six-month-accumulated precipitation shortages were persistently associated with elevated relative risk of dengue transmission in the studied period, indicative of drought impacts manifesting themselves over a sustained period of several months. Combining the results for the extreme values of the SPI-6 (dry and wet conditions), we deduce a natural interplay between drought and subsequent rainfall. In particular, whilst drought conditions during our studied period may have promoted favourable conditions for larvae habitat development (e.g. newly-introduced water storage containers), habitat suitability can only be achieved in the presence of subsequent sufficient precipitation to enable the aquatic life cycle phase of the mosquito [45]. In arid climates such as Kenya, it is estimated that extreme flooding events have increased the egg and adult abundance [46]. The longer time lags between drought conditions and heightened dengue incidence risk may also be a result of the increased survival capacity of *Ae. aegypti* eggs in dry conditions (up to 120 days) and/or more gradual changes to water storage practices [1, 12].

With respect to our model's final climatic covariate, associations between the ENSO and dengue outbreaks have been well-established for many years, as warming El Niño and cooling La Niña events can produce substantial anomalies in both precipitation and temperatures (which are conducive to mosquito development and dengue transmission) [13, 47]. Here, we contributed to the existing evidence base surrounding the effects of ENSO on incidence of new dengue cases in the Peruvian coastal region. We uncover that the coastal-specific ICEN E-Index shared stronger associations than the ONI (which is measured for the entire Niño 3·4 region) and that large-magnitude values of the ICEN E-Index, lagged by one to four months, induced elevated risk of incidence of new dengue cases [30, 48]. We identified that; i) larger positive values of the ICEN E-Index (reflective of strong El Niño events), at lags beyond two months, produced higher risk of incidence of new dengue cases, possibly due to warmer El Niño-induced ambient temperatures and drought conditions (although heavy rainfall events can also be induced by El Niño), and ii) more extreme negative values of the E-Index in the preceding one to two months (reflective of La Niña events) were also associated with greater risk of incidence of new dengue cases (albeit with smaller absolute effects), presumably due to the resulting extremely wet conditions. The findings display that lagged effects are particularly important for modelling incidence of new dengue cases in the coastal region, as SST anomalies naturally take time to culminate in extreme meteorological events onshore. These discoveries also reinforce our finding that risk induced by drought conditions (typically from El Niño events) operated at longer time lags than extreme rainfall (typically from La Niña events), and when restricting ourselves to the Peruvian coast, El Niño events are more pertinent (compared to La Niña events) for modelling incidence of new dengue cases.

Our analysis is subject to limitations. First, the reliability of our inferences is dependent on the quality of the climate and surveillance data used. For example, due to the dengue surveillance data available for Peru, we cannot directly account for individual serotypes of DENV (as two serotypes are known to circulate in the studied departments; DENV1 and DENV2) or variable reporting rates (both in time and space). The reliability of surveillance data may vary during the studied period. For instance, under-reporting may have resulted from events such as the damage caused to 299 health care centres in Piura during the 2016–2017 ENSO, which could have been additionally confounded by increased notification of other infectious diseases (such as Zika and leptospirosis) [49]. Such variability in reporting rates within and across regions and years may have impacted the model-fitting process, and yielded substantial over-dispersion and/or artificially affected accuracy measures. We cannot exclude the possibility of additional unobserved confounders in the form of socioeconomic conditions such as population density, clean water supply, living conditions, education level, sanitation infrastructure, and access to healthcare [39, 50–52]. For example, poor access to water supplies promotes

storage in potable water containers, particularly during drought conditions, thus stimulating mosquito abundance [53]. In contrast, access to private water wells and health education status are negatively associated with indices of larval abundance [54]. Similarly, we do not directly estimate the time- or space-varying risks induced by climatic influences. Thus, we indirectly adjusted for such unobserved confounders in our Bayesian spatiotemporal model (Section 2.3) via unstructured spatial effects, whilst we included yearly random effects to adjust for plausible year-to-year heterogeneities (such as changes in reporting rates or vector control programmes).

Alongside our investigation of climatic influences over time, we have presented an important use case for our climate-based model framework; reliable, robust forecasting of a subsequent month's incidence of new dengue cases. For forecasting of outbreak events, we report strong forecasting skill, including detection of 100% of all outbreaks with DIR $\geq$ 50 (or 150) per 100,000 one month ahead of time, and a corresponding AUC of 0.94 (0.97). We note two caveats and thus, potential for future model improvements; i) DIR observations with values marginally above the defined outbreak thresholds possessed low forecasted model-based posterior probabilities of breaching thresholds, and ii) due to the dependence on autocorrelation effects and a momentum oscillator, our model can currently only forecast incidence of new dengue cases with one month forecast horizons. Future work may aim to develop and implement a forecasting model with greater forecast horizons, include additional socioeconomic variables and variability in climatic conditions (e.g. diurnal temperature range), and/or refocus on DIR at a district or province level (though higher resolution data could present additional challenges e.g. spatial heterogeneities in reporting rates).

Nevertheless, using our current modelling framework, we produced representative estimates of future trends in new dengue cases and reliable classifications of future outbreaks (with robust detection of future public health crises and emergencies). Thus, the modelling framework and associated forecasts could be readily integrated into future early warning systems in Peru, and assist in the well-timed deployment of vector control resources and information campaigns. Surveillance and response strategies are coordinated by the Ministry of Health in partnership with regional (coarser spatial resolution than departments) health authorities [18]. Thus, existing strategies (e.g. targeted larvicidal treatments of standing water and insecticide spraying) could be enhanced in their timing, effectiveness, and efficiency. Due to low probabilities of false alarms, public health authorities could have a high degree of confidence in the extensive capital and resource investment involved in such vector control campaigns. Such targeted campaigns could help to control the size of future outbreaks, reduce healthcare pressures, and potentially lower dengue-related morbidity and mortality. Hence, we conclude that contingent on the timely availability of high-quality surveillance data and climate data, authorities could employ our interpretable, climate-based modelling framework to i) model and forecast current and future trends respectively in incidence of new dengue cases and ii) quantitatively inform public health decision-making.

## Supporting information

**S1 Appendix.** Table A: Summary of acronyms used throughout the analysis. Table B: Summary statistics for the studied Peruvian departments. Population refers to the last mid-year population estimate of a department's population within the studied period, as of 30 June 2021, whilst Population Density is the corresponding estimated population per km2. DIR is the mean dengue incidence rate per 100,000 across the studied period of 2010 to 2021. Maximum Temperature and Minimum Temperature similarly relate to the mean average observed value of the monthly means of maximum daily temperatures and of the monthly means of

minimum daily temperatures. Total Precipitation is the mean of monthly observed total precipitations. Table C: Summary statistics for LOOCV outbreak detection performance. Out-of-sample observations ($n$ = 417), across 2010 to 2021, were classified as being predicted true outbreaks such that DIR exceeded 50 (or 150) per 100,000 if the posterior probability of DIR exceeding 50 (or 150) per 100,000 was greater than a calibrated cut-off probability of 0.16 (or 0.13). True Positive measures the hit rate, or equivalently the proportion of true outbreaks correctly detected, whilst False Positive measures the proportion of non-outbreaks incorrectly classified as outbreaks. Accuracy represents the proportion of observations whose outbreak classification matched the true observed state (such as a predicted outbreak coinciding with an outbreak). Precision measures the proportion of predicted outbreaks which were true outbreaks. Finally, the AUC is the area under the Receiver Operating Characteristic (ROC) curve, is used as a measure of our model's skill for distinguishing between outbreaks and non-outbreaks, and here, represents an upper bound on the model's retrospective performance (due to calibration of our cut-off probability for outbreak classification). 95% confidence intervals (CIs) for summary statistics were derived using exact binomial confidence limits, with the exception of the 95% CI for AUC being calculated using 2,000 bootstrap replicates. Table D: Summary statistics for non-model-based forecasting outbreak detection across 2018 to 2021. DIR (Dengue Incidence Rate) observations ($n$ = 180) were classified one month ahead of time as being predicted true outbreaks with DIR exceeding 50 (or 150) per 100,000 only if the previous month's DIR had exceeded 50 (or 150) per 100,000. The performance metrics below, and associated 95% confidence intervals (CIs), each have the same interpretation as defined in Table C. Classification of outbreaks below can be compared to our model-based classification of future outbreaks in Table 1, where 100Fig A: Exploratory analysis of climate-dengue relationships. Cross-correlation plots for monthly lags of six climatic variables and the monthly dengue incidence rate (DIR) per 100,000 in each of the three studied departments. The plots depict a strong degree of shared similarity across departments in the timing and nature of relationship between the climate exposure and DIR. Negative lags correspond to lead times in months. Fig B: Omitting RSI and seasonality effects, exposure-lag-response relationships between climatic variables and dengue incidence. Plots of relative risk (RR), on a logarithm scale, for the included DLNMs (Distributed Lag Non-linear Models) in a climate-based Bayesian hierarchical model for dengue incidence fitted to the entire period of 140 months, where RR is defined relative to the risk induced by the mean observed value of each climate variable. Log RR values greater than 0 (pink to purple) correspond to heightened relative risk of dengue incidence, whilst values less than 0 (green) correspond to reduced RR. The four climatic variables were included in the model via DLNM specifications alongside spatial random effects, temporal random effects, and spatiotemporal random effects, but not including our usual fixed effects (of momentum and seasonality). Fig C: Omitting maximum temperature effects, exposure-lag-response relationships between climatic variables and dengue incidence. Plots of relative risk (RR), on a logarithm scale, for the included DLNMs (Distributed Lag Non-linear Models) in a climate-based Bayesian hierarchical model for dengue incidence fitted to the entire period of 140 months, where RR is defined relative to the risk induced by the mean observed value of each climate variable. Log RR values greater than 0 (pink to purple) correspond to heightened relative risk of dengue incidence, whilst values less than 0 (green) correspond to reduced RR. The three climatic variables were included in the model via DLNM specifications alongside spatial random effects, temporal random effects, and spatiotemporal random effects, and fixed effects (of momentum and seasonality). Fig D: Omitting precipitation effects, exposure-lag-response relationships between climatic variables and dengue incidence. Plots of relative risk (RR), on a logarithm scale, for the included DLNMs (Distributed Lag Non-linear Models) in a climate-based Bayesian hierarchical model for dengue incidence

fitted to the entire period of 140 months, where RR is defined relative to the risk induced by the mean observed value of each climate variable. Log RR values greater than 0 (pink to purple) correspond to heightened relative risk of dengue incidence, whilst values less than 0 (green) correspond to reduced RR. The three climatic variables were included in the model via DLNM specifications alongside spatial random effects, temporal random effects, and spatiotemporal random effects, and fixed effects (of momentum and seasonality). Fig E: Omitting the drought indicator, exposure-lag-response relationships between climatic variables and dengue incidence. Plots of relative risk (RR), on a logarithm scale, for the included DLNMs (Distributed Lag Non-linear Models) in a climate-based Bayesian hierarchical model for dengue incidence fitted to the entire period of 140 months, where RR is defined relative to the risk induced by the mean observed value of each climate variable. Log RR values greater than 0 (pink to purple) correspond to heightened relative risk of dengue incidence, whilst values less than 0 (green) correspond to reduced RR. The three climatic variables were included in the model via DLNM specifications alongside spatial random effects, temporal random effects, and spatiotemporal random effects, and fixed effects (of momentum and seasonality). Fig F: Omitting El Niño effects, exposure-lag-response relationships between climatic variables and dengue incidence. Plots of relative risk (RR), on a logarithm scale, for the included DLNMs (Distributed Lag Non-linear Models) in a climate-based Bayesian hierarchical model for dengue incidence fitted to the entire period of 140 months, where RR is defined relative to the risk induced by the mean observed value of each climate variable. Log RR values greater than 0 (pink to purple) correspond to heightened relative risk of dengue incidence, whilst values less than 0 (green) correspond to reduced RR. The three climatic variables were included in the model via DLNM specifications alongside spatial random effects, temporal random effects, and spatiotemporal random effects, and fixed effects (of momentum and seasonality). Fig G: Historical climatic time series. Exploratory plots which depict the monthly climatic data in Piura, Tumbes and Lambayeque over 140 months from May 2010 to December 2021. Both indicators for El Niño events (ICEN and ONI) are common across the three departments. Fig H: Department-specific seasonality. A decomposition of the seasonality component in dengue incidence rates in each of the departments, calculated using the bfast package in R, which iteratively decomposes a time series into trend and seasonality components whilst detecting abrupt changes within the components. The results of the bfast algorithm indicate similarity in the seasonal trends in dengue incidence rates across each of the three departments. Fig I: Cumulative association between climatic variables and DIR. Plots of cumulative risk for the included DLNMs in climate-based Bayesian hierarchical model for dengue incidence, where cumulative risk is defined relative to the cumulative risk induced by the mean observed value of each climate variable. Cumulative risk values greater than 1 correspond to heightened cumulative risk of dengue incidence, whilst values less than 1 correspond to reduced cumulative risk. For example, as cumulative risk was negative for a monthly precipitation of 1mm at the current lag, then exposure over one month to dry conditions was associated with reduced risk, whilst positive values of cumulative risk at two months indicates that the exposure over two months resulted in a increased cumulative risk in dengue incidence. Note that in each sub-plot, we define a high and low value of the climatic variable, and the cumulative risk at each lag is defined relative to the cumulative risk induced by the mean value of the climatic variable. Fig J: In-sample predictive performance of our Bayesian spatiotemporal model. The DIR (Dengue Incidence Rate) time series (gold) for each department is plotted alongside the posterior median estimate (forest green) for each observation and the corresponding estimated 95% credible intervals (shaded grey). Fig 4 is the corresponding figure for the cross-validation setting which display the estimated out-of-sample predictive performance and thus, approximates the model's generalisability. Fig K: Estimated posterior distributions of department-specific yearly random

effects. The visualization displays the posterior median estimate (circles) of the effect size (on a logarithmic scale) of the yearly random effect for each department, whilst the errorbars capture the 95% credible intervals. Note that the effect size can be interpreted as the contribution of the department-specific yearly random effect to the logarithm of DIR (Dengue Incidence Rate) observations within each year. The variability across departments in the yearly random effects is indicative of the differences in trends of year-to-year heterogeneities which prevented a shared yearly random effect (for each department) providing sufficient explanatory power. Fig L: Estimated posterior distributions of department-specific monthly random effects. The visualization displays the posterior median estimate (circles) of the effect size (on a logarithmic scale) of the monthly random effect for each department, whilst the errorbars capture the 95% credible intervals. Note that the effect size can be interpreted as the contribution of the department-specific monthly random effect to the logarithm of DIR (Dengue Incidence Rate) observations across the years. Note that the contribution to DIR is in addition to the common seasonality indicator variable for summer months. Fig M: Estimated posterior distributions of department-specific structured spatial component effects. The visualization displays the posterior median estimate (circles) of the effect size (on a logarithmic scale) of the spatially structured random effect component of the BYM2 model for each department, whilst the errorbars capture the corresponding 95% credible intervals. Note that the effect size can be interpreted as the contribution of the department-specific spatial random effect to the logarithm of DIR (Dengue Incidence Rate) observations across the years. Fig N: Estimated posterior distributions effects of momentum indicator. The visualization displays the posterior distribution of the effect size (on a logarithmic scale) of the lagged momentum indicator. Note that the effect size can be interpreted as the contribution of the lagged momentum indicator, calculated for three months of DIR (Dengue Incidence Rate) observations, to the current logarithm of DIR. We use the lagged momentum indicator as this ensures that the covariate does not include the response (DIR) itself. Fig O: Leave-one-out cross-validation errors visualization. The histogram depicts the posterior median estimate of the absolute error for each observation, which was derived individually for observations by computing the median absolute difference between the posterior predictive value and the corresponding observed DIR (Dengue Incidence Rate). 89.6% of the posterior median estimates ($n = 180$) for the absolute forecast errors are less than 50 per 100,000. Fig P: Retrospective modeling outbreak detection. Among the observations with DIR (Dengue Incidence Rate) greater than 50 (left) and greater than 150 (right) per 100,000, the plots depict posterior probabilities of DIR exceeding thresholds (green) of 50 and 150 respectively, obtained via a leave-one-time-point-out fitting of models. The plots capture the model framework's sensitivity in estimating substantial out-of-sample dengue outbreaks. Within the two subsets of observations, the median posterior probabilities of exceeding 50 per 100,000 and 150 per 100,000 are 0·68 and 0·57 respectively. To also display the false positive rate and thus ensure a representative picture of our model's outbreak detection capabilities, Fig Q is the analogous visualization for the corresponding posterior probabilities of the observations with DIR less than the thresholds of 50 and 150 per 100,000. Fig Q: Reliability of estimated out-of-sample outbreaks. Among the observations with DIR (Dengue Incidence Rate) less than 50 (left) and DIR less than 100 (right), the plots depict posterior probabilities of forecasted DIR exceeding thresholds of 50 per 100,000 (left) and 150 per 100,000 (right). The plots visualize the model's false positive rate in terms of forecasting severe dengue outbreaks. In the subset of observations with DIR less than 50, the mean posterior probability of exceeding 50 is 0·05, whilst the maximum posterior probability of exceeding 50 is 0·60. Similarly, in the subset of observations with DIR less than 150, the mean posterior probability of exceeding 50 is 0·08, whilst the maximum posterior probability of exceeding 50 is 0·43. In terms of the true positive rate (or hit rate), Fig P is the analogous visualization for

the corresponding posterior probabilities of the observations with DIR greater than the thresholds of 50 and 150. Fig R: Forecasting errors visualization. The histogram depicts the posterior median estimate of the absolute error for each observation, which was derived individually for observations by computing the median absolute difference between the posterior predictive value and the corresponding observed DIR. 89.2% of the posterior median estimates ($n = 180$) for the absolute forecast errors are less than 50 per 100,000. Fig S: Reliability of forecasted outbreak detections. Among the observations with DIR less than 50 (left) and DIR less than 100 (right), the plots depict posterior probabilities of forecasted DIR exceeding thresholds of 50 per 100,000 (left) and 150 per 100,000 (right). The plots visualize the model's false positive rate in terms of forecasting severe dengue outbreaks. In the subset of observations with DIR less than 50, the mean posterior probability of exceeding 50 is 0·05, whilst the maximum posterior probability of exceeding 50 is 0·59. Similarly, in the subset of observations with DIR less than 150, the mean posterior probability of exceeding 150 is 0·02, whilst the maximum posterior probability of exceeding 50 is 0·43. In terms of the true positive rate (or hit rate), Fig 6 is the analogous visualization for the corresponding posterior probabilities of the observations with DIR greater than the thresholds of 50 and 150.
(PDF)

## Author Contributions

**Conceptualization:** Cathal Mills.

**Data curation:** Cathal Mills.

**Formal analysis:** Cathal Mills.

**Investigation:** Cathal Mills.

**Methodology:** Cathal Mills, Christl A. Donnelly.

**Software:** Cathal Mills.

**Supervision:** Christl A. Donnelly.

**Validation:** Cathal Mills.

**Visualization:** Cathal Mills.

**Writing – original draft:** Cathal Mills.

**Writing – review & editing:** Cathal Mills, Christl A. Donnelly.

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
