## [Decision Letter · Decision Letter 0]

3 Jun 2024

Dear Mr Mills,

Thank you very much for submitting your manuscript "Climate-based modelling and forecasting of dengue fever in three endemic departments of Peru" for consideration at PLOS Neglected Tropical Diseases. As with all papers reviewed by the journal, your manuscript was reviewed by members of the editorial board and by independent reviewers. The reviewers appreciated the attention to an important topic. Based on the reviews, we are likely to accept this manuscript for publication, providing that you modify the manuscript according to the review recommendations. 

Sincerely,

Christopher M. Barker

Academic Editor

Andrea Marzi

Section Editor

Reviewer's Responses to Questions

**Key Review Criteria Required for Acceptance?**

**Methods**

-Are the objectives of the study clearly articulated with a clear testable hypothesis stated?

-Is the study design appropriate to address the stated objectives?

-Is the population clearly described and appropriate for the hypothesis being tested?

-Is the sample size sufficient to ensure adequate power to address the hypothesis being tested?

-Were correct statistical analysis used to support conclusions?

-Are there concerns about ethical or regulatory requirements being met?

Reviewer #1: All YES

Reviewer #2: The study's objectives are clearly articulated, focusing on developing a climate-based Bayesian spatiotemporal model to understand and predict dengue incidence in specific regions of Peru. The hypothesis is implicit, proposing that incorporating climatic variables and specific indicators will improve dengue incidence predictions. The study design is appropriate, employing a bespoke Bayesian spatiotemporal model that integrates region-specific climatic knowledge and dengue incidence data. The design includes model fitting, retrospective modelling, and real-time forecasting, all of which align with the study's objectives.

The population is clearly described, focusing on three Peruvian departments: Piura, Tumbes, and Lambayeque. This regional focus is appropriate given the hypothesis that climate impacts dengue incidence differently in various areas. The sample size of 140 months of dengue notifications across three departments appears sufficient for the study's complex modelling framework. The use of cross-validation methods further strengthens the robustness of the analysis.

The study employs advanced statistical methods, including a Bayesian hierarchical model, negative binomial likelihood, and Integrated Nested Laplace Approximation (INLA). These methods are appropriate for handling spatiotemporal data and overdispersion, supporting the study's conclusions effectively. The summary does not explicitly address ethical or regulatory concerns. However, as it involves epidemiological data and statistical modelling rather than direct human or animal subjects, the primary ethical considerations would involve data privacy and accurate reporting, which are not indicated as problematic here.

Overall, the study is well-designed with a robust statistical framework, addressing the objectives through comprehensive modelling and validation techniques. The population is well-chosen, and the sample size is adequate for the study's aims. The statistical analyses are correctly applied, ensuring the conclusions are well-supported. While not explicitly discussed, there are no apparent ethical concerns given the nature of the study. The study appears thorough, well-structured, and methodologically sound, capable of providing valuable insights into the relationship between climate variables and dengue incidence in the specified regions of Peru.

**Results**

-Does the analysis presented match the analysis plan?

-Are the results clearly and completely presented?

-Are the figures (Tables, Images) of sufficient quality for clarity?

Reviewer #1: Generally YES. As mentioned in my review it would be nice to see the dengue data

Reviewer #2: The analysis presented closely matches the analysis plan, as it follows a clear and systematic approach to examining the relationship between climatic factors and dengue incidence. The results are clearly and comprehensively presented, detailing the relationships identified between climate variables and dengue incidence rates (DIRs) at various lag times. The use of exploratory cross-correlation plots (Figure 2) effectively captures the structure and intensity of these relationships, demonstrating strong similarities across different departments and indicating potential for useful climate-based forecasts at least one month ahead.

The Bayesian spatiotemporal model framework, including distributed lag non-linear models (DLNMs), precisely describes the relationships between DIRs and climatic variables while accounting for seasonality, momentum effects, and both temporal and spatial random effects. The results, including three-dimensional exposure-lag-response relative risk (RR) plots (Figure 3), show clear patterns: higher maximum temperatures and extreme precipitation conditions are associated with elevated dengue risk. Additionally, the study finds that both moderate-to-extreme rainfall and severe drought, as measured by the SPI-6 index, influence dengue incidence, and that El Niño events significantly impact RR.

The figures and tables presented are of high quality and provide sufficient clarity. Figures like the cumulative risk plots and exposure-lag-response plots effectively illustrate the findings. Supplementary materials further enhance understanding by providing detailed visualizations of model performance and error distributions.

The model’s performance was validated through various metrics, such as Deviance Information Criterion (DIC), cross-validated (CV) log score, Mean Absolute Error (MAE), and cross-validated posterior predictive checking. The LOOCV posterior predictive check revealed that 94.7% of DIR observations fell within the 95% credible intervals, demonstrating the model's robustness. Visual assessments of out-of-sample predictive performance (Figure 4) and model-based errors (Supplementary Material Figure SI 13) indicate a moderate-to-strong ability to track dengue trends, despite occasional under-estimation of extreme outbreaks.

The model’s capability to detect outbreaks was further assessed through probabilistic estimates, showing high accuracy in classifying out-of-sample outbreaks with thresholds of 50 and 150 per 100,000 DIR. The false positive rates were low, indicating reliable outbreak detection.

In the forecasting analysis, the model accurately predicted subsequent month’s DIRs from 2018 to 2021, with 93.9% of DIR observations within the 95% credible intervals. The model successfully anticipated future dengue outbreaks, achieving 100% correct forecasts for both outbreak thresholds with relatively low false positive rates. Figure 6 and Supplementary Material Figure SI 15 effectively visualize the correspondence between observed and forecasted DIR values, showcasing the model's forecasting reliability. Overall, the analysis is thorough and the presentation of results is clear, supported by high-quality figures and rigorous validation methods.

**Conclusions**

-Are the conclusions supported by the data presented?

-Are the limitations of analysis clearly described?

-Do the authors discuss how these data can be helpful to advance our understanding of the topic under study?

-Is public health relevance addressed?

Reviewer #1: All YES

Reviewer #2: The conclusions are strongly supported by the data, as the climate-based Bayesian hierarchical model accurately captured spatiotemporal trends in dengue incidence. Extensive model validation confirmed the reliability of the findings.

The analysis acknowledges limitations, such as the quality of climate and surveillance data, variability in dengue incidence reporting, and the absence of socioeconomic confounders. These are addressed by including unstructured spatial effects and yearly random effects.

The study advances understanding by identifying specific climatic influences on dengue risk and their lagged effects, corroborated by existing research.

Public health relevance is highlighted through the model’s ability to forecast dengue outbreaks one month ahead with high accuracy. This can inform early warning systems, aiding timely vector control measures and reducing dengue-related healthcare pressures. The study's findings are thus valuable for public health decision-making.

**Editorial and Data Presentation Modifications?**

Reviewer #1: Minor revision. There are a number of aspects that should be relatively easy to add/change and which would improve the paper. These are in my attached word doc.

Reviewer #2: The paper is written very well.

**Summary and General Comments**

Reviewer #1: This is a very good analysis with astonishingly good results for predicting dengue epidemics with only 12% false positive rates. What is lacking is more information on the three study sites and the acquired dengue data, as well as a number of smaller points throughout that need some clarification.

Reviewer #2: I thank the authors for sharing this insightful document, which presents a robust analysis of dengue incidence using a climate-based Bayesian hierarchical model. The study accurately captures spatiotemporal trends and has strong forecasting abilities, predicting dengue outbreaks one month in advance with high reliability. It enhances our understanding of the relationship between climatic variables and dengue risk and has clear public health relevance, aiding in timely deployment of vector control resources.

PLOS authors have the option to publish the peer review history of their article (what does this mean?). If published, this will include your full peer review and any attached files.

Reviewer #1: No

Reviewer #2: No

Figure Files:

Data Requirements:

Reproducibility:

References

---

## [Decision Letter · Decision Letter 1]

2 Oct 2024

Dear Mr Mills,

We are pleased to inform you that your manuscript 'Climate-based modelling and forecasting of dengue fever in three endemic departments of Peru' has been provisionally accepted for publication in PLOS Neglected Tropical Diseases.

**A reviewer (and we as editors) have not been able to locate the data that are stated to be publicly available "from the National Centre for Epidemiology, Disease Prevention and Control (Peru CDC) in Peru’s Ministry of Health at " ext-link-type="uri" xlink:type="simple">https://www.dge.gob.pe/." The link provided is to a general website that leaves a lot of work to the reader to locate the data. To avoid any further delays, we are accepting the manuscript instead of requesting minor revision, but we ask that you please include a clearer definition of the data source, including exactly how and where one might locate the public data used in your analyses.**

IMPORTANT: The editorial review process is now complete. PLOS will only permit corrections to spelling, formatting or significant scientific errors from this point onwards (except the requested clarification regarding the public data source above). Requests for major changes, or any which affect the scientific understanding of your work, will cause delays to the publication date of your manuscript.

Best regards,

Christopher M. Barker

Academic Editor

Andrea Marzi

Section Editor

Please consider moderating claims about the high degree of novelty or pioneering nature of this paper, as some seem exaggerated. Incremental progress has value, and we agree that this study is a useful advance. Modeling and prediction of dengue at various spatio-temporal scales with a wide range of model forms, including zero-inflated and machine learning models, have been published previously. It is somewhat misleading to claim that "this is the first instance in which dengue prediction accuracy has been compared across various spatiotemporal resolutions." Even in the unlikely event that this is true, many other studies on dengue prediction have collectively evaluated predictive performance across a very wide range of spatio-temporal resolutions.

We appreciate your response to the reviewers' concerns, and the reviewers have provided feedback on the revised manuscript.

Reviewer's Responses to Questions

**Key Review Criteria Required for Acceptance?**

**Methods**

-Are the objectives of the study clearly articulated with a clear testable hypothesis stated?

-Is the study design appropriate to address the stated objectives?

-Is the population clearly described and appropriate for the hypothesis being tested?

-Is the sample size sufficient to ensure adequate power to address the hypothesis being tested?

-Were correct statistical analysis used to support conclusions?

-Are there concerns about ethical or regulatory requirements being met?

Reviewer #1: I thank the authors for taking on board my comments. It reads very well now.

**Results**

-Does the analysis presented match the analysis plan?

-Are the results clearly and completely presented?

-Are the figures (Tables, Images) of sufficient quality for clarity?

Reviewer #1: I thank the authors for taking on board my comments. It reads very well now. The results are excellent.

**Conclusions**

-Are the conclusions supported by the data presented?

-Are the limitations of analysis clearly described?

-Do the authors discuss how these data can be helpful to advance our understanding of the topic under study?

-Is public health relevance addressed?

Reviewer #1: I thank the authors for taking on board my comments. It reads very well now.

**Editorial and Data Presentation Modifications?**

Reviewer #1: (No Response)

**Summary and General Comments**

Reviewer #1: I thank the authors for taking on board my comments. It reads very well now.

PLOS authors have the option to publish the peer review history of their article (what does this mean?). If published, this will include your full peer review and any attached files.

Reviewer #1: No

---

## [Editor Report · Acceptance letter]

15 Nov 2024

Dear Mr Mills,

We are delighted to inform you that your manuscript, "Climate-based modelling and forecasting of dengue in three endemic departments of Peru," has been formally accepted for publication in PLOS Neglected Tropical Diseases.

Best regards,

Shaden Kamhawi

co-Editor-in-Chief

Paul Brindley

co-Editor-in-Chief
